# Publication bias and the tourism-led growth hypothesis

**Nikeel Nishkar Kumar** [1,2]* , **Arvind Patel**[1] , **Ravinay Amit Chandra**[3‡], **Navneet Nimesh Kumar**[4‡]

1 School of Accounting, Finance, and Economics, The University of the South Pacific, Suva, Fiji Islands, 2 School of Economics, Faculty of Business, Economics and Law, Auckland University of Technology, Auckland, New Zealand, 3 Department of Management, School of Business and Economics, The University of Fiji, Lautoka, Fiji, 4 School of Business and Management, The University of the South Pacific, Suva, Fiji Islands

☯ These authors contributed equally to this work.
‡ These authors also contributed equally to this work.
* nikeelkumar3@gmail.com, nikeel.kumar@usp.ac.fj, nikeel.kumar@autuni.ac.nz, nikeel.kumar@aut.ac.nz

## Abstract

This study attempts to solve the publication bias suggested by recent review articles in the tourism-growth literature. Publication bias is the tendency to report favourable and significant results. Method and data triangulation, and the Solow-Swan model are applied. A sample from 1995 to 2018 is considered with Tonga as a case study. The approach consists of multiple methods, data frequencies, exchange rates, structural breaks, and an overall tourism index developed using principal component analysis (PCA). Consistent results across these dimensions are obtained with the PCA models. Tourism has small, positive, and statistically significant economic growth effects. Theoretically consistent values of the capital share and exchange rates are obtained. The results indicate the importance of multiple methods and the overall tourism index in assessing the tourism-growth relationship and minimising publication biases. The practical implication is the provision of robust elasticity estimates and better economic policies.

## Introduction

International tourism is generally considered a key sector for growth and development in developing countries. It attracts foreign exchange and allows developing countries to import new capital goods [1]. Imported capital goods are infused with new technology which raises the technology level in the country. It supports human capital accumulation as workers acquire new skills and knowledge by using the new capital goods [1]. Apart from productivity gains, there are spill-over benefits because the new skills and knowledge circulates freely between industries. In contrast, tourism is associated with negative externalities such as environmental degradation and the overutilization of natural resources [2]. On balance however, research suggests that developing the tourism industry is supportive of economic growth [3].

Yet, recent meta-analyses by Nunkoo et al. [3] and Fonseca and Sánchez-Rivero [4, 5] raise concerns over the issue of publication bias in empirical research on tourism and

**Competing interests:** The authors have declared that no competing interests exist.

economic growth. Publication biases occur due to the preferential reporting of positive and statistically significant results [6]. Two types of publication bias are reported, types I and II [3]. Under type I bias, researchers tend to report strongly positive/negative estimates [7, 8]. Under type II bias, researchers tend to report significant, yet economically meaningless results [3]. The non-reporting of inconclusive results, coupled with the publication of statistically significant but economically meaningless results leads to biased estimates, distorted inference, and skewed knowledge, which undermine the free exchange of information [3]. Publication biases result in overoptimistic inferences about an economic phenomenon such as the contribution of tourism to growth [3–5]. As a result, the belief in the efficacy of tourism policies may be unfounded or that policies may have a smaller than expected effect on economic growth [3].

The usual method to detect publication bias begins with a funnel asymmetry plot where the empirical effects are plotted against the inverse of the standard error of the estimates [4]. Without publication bias, the funnel graph tends to be symmetric and resemble an inverted funnel. Asymmetry in the funnel graph provides preliminary evidence on publication biases which is formally tested through meta-regression analysis (MRA) [9]. Type I bias is tested for by regressing the $t$-statistic against the inverse of the standard error of the empirical effect under analysis [4, 5]. Type II bias is similarly tested except the absolute value of the $t$-statistics, instead of the $t$-statistic, is considered [4, 5]. Types I and II publication biases are confirmed if the MRA intercept term is significantly greater than zero. Tourism genuinely affects growth if the slope of the standard error term in the MRA is significantly different from zero. These are termed the funnel asymmetry and precision effects tests, respectively [4, 5].

By surveying 545 estimates across 113 studies published from 1994 to 2017, Nunkoo et al. [3] conclude that positive and significant effects are preferentially reported in the tourism-growth literature. Fonseca and Sánchez-Rivero [4] also confirm that the results reported in the tourism-growth association are non-genuine, but the variability of the empirical effects depends on the degree of tourism specialization, level of economic development, and size of the countries analysed. Fonseca and Sánchez-Rivero [5] further note that statistical significance may occur in small samples if econometric specifications are manipulated to find larger effects. This indicates type II bias because small samples are associated with larger standard errors and smaller effect estimates [5]. Overall, the effects are sensitive to specification and estimation characteristics, data frequency, and period considered [3–5].

To solve these problems, Nunkoo et al. [3] recommends re-visiting the tourism-growth association across various methodological characteristics, specifications, and estimation choices. Song and Wu [10] emphasize that research designs that ignore theoretical foundations that underpin the tourism-growth nexus lead to unreliable and misleading conclusions [10]. Song and Wu [10] underscore the Solow-Swan growth model and accentuate how tourism improves technological progress and productivity of capital and labor. To measure tourism, Shahzad et al. [11] suggest that combining multiple indicators such as arrivals and receipts into a composite index captures information on the traditional variables and is less affected by multicollinearity [11]. Solarin [12] notes that ignoring exchange rates in the specification may bias the growth effect of tourism. Controlling for structural breaks is also important because tourism is impacted by exogenous events [3]. On estimation, dynamic methods such as ARDL derive more accurate conclusions on the validity of the tourism-growth association [3].

Consequently, the objective of this study is to remedy publication biases in the tourism-growth literature. Method and data triangulation are adopted, and various methodological characteristics, specifications, and estimation choices are considered [13]. The earliest

application of triangulation in the social sciences was by Eugene et al. [14] and its primary rationale is the recognition of data-set or methodological biases with a single method or data-set [15]. Triangulation is the use of multiple approaches to a research question enabling the researcher to "zero in" on the answers sought [13]. Method triangulation is the use of more than one research method in measuring the object of interest with a data set [13]. Data triangulation refers to using the same approach/method with different sets of data to verify/falsify the generalizable trends observed in one dataset [13].

For method triangulation, the autoregressive distributed lag (ARDL) [16], dynamic least squares (DOLS) [17], fully modified least squares (FMOLS) [18], and canonical cointegrating regression (CCR) [19, 20] are used. These methods are consistent with cointegration theory, and provide robust estimates in the presence of endogeneity and small samples. Endogeneity and small samples can bias the effect estimates. Small samples can also inflate standard errors which may lead to type II publication bias [5]. However, using multiple methods with a single dataset makes the results more reliable by lowering the likelihood that econometric techniques are manipulated. The primary long-run results are supported with the ridge regression technique which controls for multicollinearity [21–23].

For data triangulation, three measures of tourism are considered. These are tourist arrivals, tourism receipts, and an overall tourism index which is developed using the Principal Component Analysis (PCA) technique. Because the results are affected by the frequency of the data, annual and quarterly data series are also considered. Shahzad et al. [11] and Shahbaz et al. [24] demonstrate the usefulness of frequency transformation techniques from low to high-frequency data in addressing the small sample bias. Shahzad et al. [11] and Shahbaz et al. [24] recommend the quadratic match-sum approach for frequency conversion. This approach is ideal in minimizing the seasonality problem by reducing point-to-point data variations and has previously been successfully applied in the tourism-growth literature [11, 24].

Tonga is chosen as a case study with an annual sample from 1995 to 2018 which amounts to 24 years of data. Like other Pacific Island Countries (PICs), international tourism is for Tonga, a key driver of growth. In Tonga, tourism contributed 12.1 percent of Gross Domestic Product (USD 52.6 million) and supported 12 percent of total employment or about 5100 jobs in 2019 [25]. International arrivals and receipts to Tonga have been consistently rising (Fig 1). There is also conflicting evidence, and dearth of country-specific studies in Tonga [26–28]. Current research shows a positive impact but differs in terms of elasticity estimates, explanatory variables, and specification. To reconcile the conflicting evidence, an overall assessment using theoretically founded models and triangulation is required.

The key methodological contribution of this paper is the development of a cohesive framework based on triangulation to remedy publication biases in the tourism-growth literature. The methodology developed draws from the recommendation of Nunkoo et al. [3], theoretical foundations from Song and Wu [10], measurement of tourism from Shahzad et al. [11], and inclusion of moderator variables from Solarin [12]. With this framework, the study provides new evidence on how tourism interacts with growth in small PICs, namely Tonga. Notably, the size and sign of the growth effect of tourism depends on the research method and the measure of tourism. Consistent results across methods and data frequencies are obtained using the overall PCA index models. The findings indicate that tourism has small but positive effects on growth, whilst structural breaks and exchange rates have negative effects on growth. Theoretically consistent values of the capital share are found. The practical implication is on better policies to promote economic growth by developing Tonga's tourism industry.

The remainder of the study is set as follows. Section 2 discusses the methodology, Section 3 presents the results, and Section 4 concludes with policy implications.

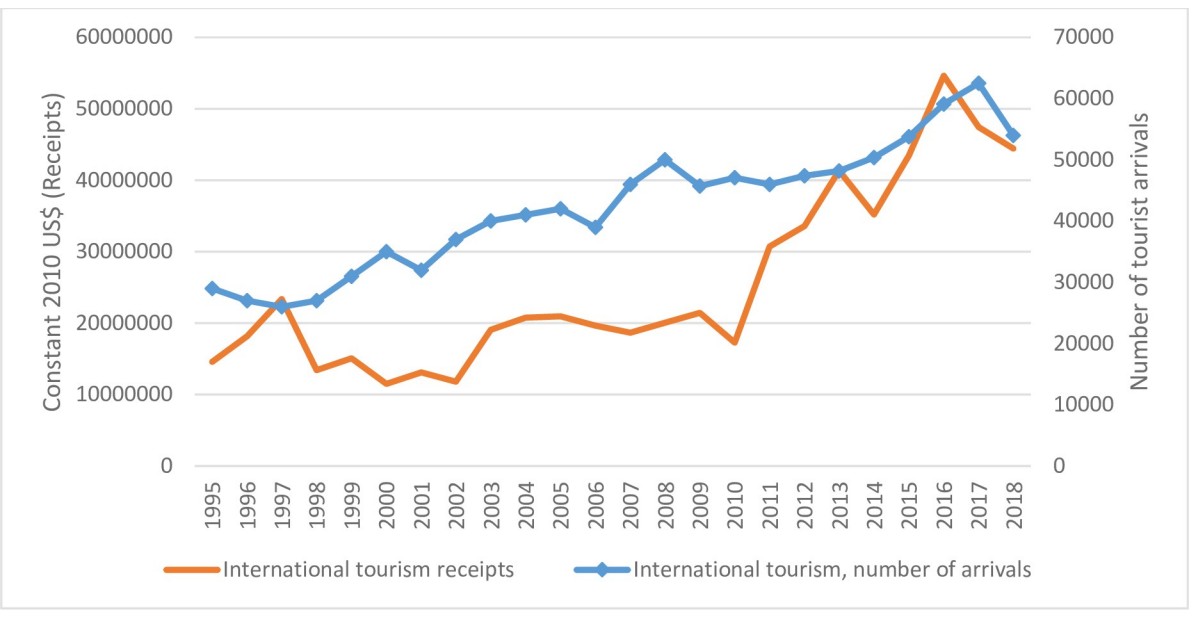

**Fig 1. Tonga's tourism sector performance—1995 to 2018.** Data obtained from World Bank.

## Materials and methods

### Theoretical model

The model specification is adapted from Solow [29]. The model begins with the Cobb-Douglas production function with capital per-worker [10]:

$$y_t = A_t k_t^{\alpha} \tag{1}$$

where y is real GDP per worker, A represents the stock of technology, k is capital per worker, and $0 < \alpha < 1$ is the capital share.

The model assumes the Hicks neutral technical progress and constant returns to scale. The evolution of technology consists of two components. Part 1 assumes that the evolution of technology is exogenous. Part 2 assumes that technical progress is a function of productivity-improving factors. Tourism and exchange rates are introduced as shift variables:

$$A_t = \underbrace{\lambda e^{gt}}_{part\ 1} \underbrace{tur_t^{\vartheta} er_t^{\eta}}_{part\ 2} \tag{2}$$

where $\lambda$ is the initial level of technology, $t$ is time, $tur_t$ is the tourism market indicator, and $er_t$ is the exchange rate. Substituting Eq (2) in place of $A_t$ in Eq (1), taking logs of the resulting equation, and including indicator variables to represent structural breaks derives the basic long-run model for estimation:

$$\ln y_t = \pi + gt + \vartheta \ln tur_t + \eta \ln er_t + \alpha \ln k_t + D + u_t \tag{3}$$

where $\pi$ is the intercept, $D$ is the structural break indicator, and $\vartheta > 0$ and $\eta > 0$. Three measures of tourism are considered. Specifically, $ta_t$ is tourist arrivals, $tr_t$ is tourism receipts, and $ti_t$ is the PCA tourism development index. Nominal and real exchange rates are considered. The exchange rates are defined with reference to the US dollar i.e. Tongan Pa'anga/USD. An increase signals a decline of the Pa'anga against the USD.

## Data

A total of 24 years of annual data over the periods from 1995 to 2018 is used for analysis. The data for real GDP in constant 2010 US dollars was available from 1981 to 2018, gross capital formation to proxy for investment was available from 1975 to 2018, tourist arrivals and tourism receipts in constant 2010 US dollars was available from 1995 to 2018, population data was available from 1960 to 2018, and the labor force participation rate was available from 1990 to 2019. The nominal exchange rate, Tongan Pa'anga against the US dollar was available from 1961 to 2020. GDP deflator for the USA and Tonga was available from 1960 to 2019, and 1981 to 2019, respectively. The average labor force participation rate was multiplied by the population to compute the labor series. The capital stock series was computed using the perpetual inventory method. The initial capital stock was set as 1.5 times the 1981 real GDP, and the depreciation rate is set at 5 percent. The real exchange rate is derived by multiplying the nominal exchange rate with the ratio of GDP deflator in the USA against Tonga. The data are sourced from the World Development Indicators and Global Development Finance database [30]. Data for gross capital formation from 2013 to 2018 is handpicked from the Asian Development Bank's Key Indicators for Asia and the Pacific Series database [31]. Data for the official exchange rate is sourced from the IMF's International Financial Statistics database [32]. Exchange rate data over the period 2014 to 2019 is sourced from the exchange rates UK website [33].

## Methods

**PCA and frequency conversion.** Principal component analysis is used to construct the overall tourism activity indicator. A key benefit of the PCA is that the resulting indicator is not subject to multicollinearity and is less sensitive to missing data in the underlying components [11]. The relevance of the underlying components is determined by the estimated eigenvalues. Following the Kaiser criterion, components with eigenvalues greater than 1 are important in the index.

Frequency conversion techniques are used to mitigate problems associated with small samples and improve degrees of freedoms [11, 24]. The quadratic match sum technique is used to convert from low to high frequencies [11, 24]. The approach fits a quadratic polynomial for each observation of the low-frequency series. The polynomial is formed via three adjacent points from the lower frequency series, and is then used to fill in observations of the higher frequency series for that period. The fitted quadratic is either the average or sum of the higher frequency points that match the lower frequency data. Points earlier and after the current period are used in the interpolation process. For endpoints, two periods earlier/after the endpoint are considered [11, 24]. This approach avoids issues related to seasonality, and the converted data are comparable to seasonally adjusted series [11, 24].

**Unit root and structural breaks.** The augmented Dickey-Fuller (ADF) [34] and Phillips-Perron (PP) [35] unit root tests are applied to examine the stationarity properties of the data. Both tests are conducted by estimating the following equation with OLS:

$$\Delta y_t = \mu_t + \varphi_1 y_{t-1} + \sum_{i=1}^{m} \Delta y_{t-i} + \epsilon_t \tag{4}$$

where $\Delta$ is the first difference operator, $y$ is a time series variable, $\mu_t$ is the deterministic component which includes intercept and/or the time trend, $\varphi_1$ is the autoregressive coefficient and $\epsilon_t$ is the error term. Lagged dependent variables are included in Eq (4) to correct for potentially auto-correlated residuals under the ADF test. The PP test corrects for auto-correlated residuals through Newey-West standard errors. The lag used in the ADF test and bandwidth in the PP tests is determined by the Schwarz information criteria. The null hypothesis in both tests is

that the underlying series has a unit root, H$_0$: $\varphi_1 = 0$. The t-statistics obtained from both methods are compared against the respective critical values. Rejection of the null hypothesis implies that the series in question is stationary.

Nonlinear flexible Fourier unit root test is applied due to potential nonlinear adjustment process [36, 37]. The nonlinear deterministic component of the data generation process is specified as follows in step one [36, 37]:

$$y_t = \alpha_0 + \alpha_1 \sin 2\pi k * t/n + \alpha_2 \cos 2\pi k * t/n + v_t \qquad (5)$$

where $y$ is a time series variable, $\pi = 3.1416$, $t$ is the time trend, $n$ is the sample size, $k^*$ is the optimal number of frequencies of the underlying Fourier function, and $v_t$ is the error term. The optimal frequency of the underlying Fourier function is obtained by assigning whole number values of $k$ changing from between 1 to 5, then estimating Eq (5) by OLS and minimizing the resulting sum of the square of the residuals [36]. In step two, the residuals of the preferred $k^*$ Eq (5) are used to test the null hypothesis of the presence of unit roots:

$$v_t = \beta_1 v_{t-1}^3 + \beta_2 v_{t-1}^2 + \sum\nolimits_{j=1}^{p} \varphi_j \Delta v_{t-j} + \varepsilon_t \qquad (6)$$

The null hypothesis is that the underlying series has a unit root, H$_0$: $\beta_1 = \beta_2 = 0$. Rejection of the null hypothesis employing an F-test implies stationarity [36]. The critical values of this test are obtained from Becker, Enders, and Lee [38]. Lags of the differenced dependent variable in Eq (6) are included up to where autocorrelation is corrected [36, 37].

Testing for structural breaks is important because tourism development is affected by exogenous events [3]. To examine the presence of structural breaks, the Bai and Perron sequential break test is used [39, 40]. The procedure is useful in that one can examine the presence of multiple structural breaks. The method produces a consistent estimation of the location and number of breaks and corrects for serial correlation across the break segments by robust standard errors.

**Cointegration, endogeneity, and small sample consistent estimates.** The ARDL bounds approach is applied to test for cointegration [16]. The benefits of this approach include the ability to examine cointegration when the series is stationary at levels, stationary at the first difference, or a combination of the two. The method provides unbiased estimates in small samples and avoids the endogeneity bias due to its dynamic structure supported by lagged explanatory variables that act as instruments in the absence of autocorrelated residuals. Lag symmetry is not required, and cointegration can be identified in multivariate systems. The ARDL model is specified as follows:

$$\Delta y_t = \mu_1 + \theta_1 y_{t-1} + \theta_2 x_{t-1} + short\ run\ terms + u_t \qquad (7)$$

where $y$ and $x$ are time series variables, $\mu_1$ is the deterministic component which includes intercept, trend, and structural breaks, and $-1 < \theta_1 < 0$ is the adjustment coefficient. The bounds test of cointegration tests whether the lagged level variables in Eq (7) estimated by OLS is significantly different from zero. Cointegration exists if the resulting F-statistic exceeds the upper critical bound, does not exist if the F-statistic is below the lower critical bound, and is inconclusive if the F-statistic falls within the upper and lower bounds [16].

Long-run estimates are also provided by the dynamic least squares (DOLS) [17], fully modified least squares (FMOLS) [18], and the canonical cointegration regression (CCR) [19, 20]. FMOLS is an optimal single-equation method-based least squares estimator with semi-parametric corrections for serial correlation and endogeneity of the explanatory variables [41]. DOLS is useful in small samples, can be applied with mixed and higher orders of integration, and can accommodate for possible simultaneity between regressors [42]. CCR approach has similar

benefits as FMOLS which is achieved by incorporating stationary components in the cointegrating models [19, 20]. The resulting CCR estimates are asymptotically efficient [19, 20].

In the final step, the average of the coefficient estimates from the alternative methods are derived, and its significance level is subsequently tested. The test statistic is computed as follows: $t^* = \dfrac{\sum \widehat{\vartheta}_i / n}{\sqrt{\sum(\widehat{\vartheta}_i - \sum \widehat{\vartheta}_i / n)^2 / (n-1)}}$ where $\widehat{\vartheta}$ are the estimated coefficients obtained from the $n = 4$ methods, $\sum \widehat{\vartheta}_i / n$ is the average of the coefficients across the alternative methods, and $\sqrt{\dfrac{\sum(\widehat{\vartheta}_i - \sum \widehat{\vartheta}_i / n)^2}{n-1}}$ is the standard deviation of the coefficient estimates from the $n = 4$ methods.

**Multicollinearity consistent estimates.** Multicollinearity is a phenomenon in which two or more predictors in multiple regression are highly correlated [23]. Strong multicollinearity creates difficulties in testing individual least squares regression coefficients due to inflated standard errors [23]. The Farrar-Glauber (FG) test is used to detect the presence of severe multicollinearity. The FG test defines multicollinearity in terms of departures from orthogonality and helps detect the presence and pattern of multicollinearity. The null hypothesis of no multicollinearity is rejected if the p-value from the test statistic is less than 5 percent.

Multicollinearity consistent estimates are achieved via ridge regression [21, 22]. This is a special case of a Tikhonov regularization where all parameters are regularized equally [23]. The approach is based on the bias-variance trade-off from machine learning models where parameter variance is reduced by marginally increasing the bias in the estimated parameters through a penalty factor. Ridge regression uses the L2 regularization technique where the squared magnitude of coefficients is added as a penalty term to the sum of squared errors:

$$\sum_{i=1}^{n} \left\{ y_i - \sum_{j=1}^{p} x_{ij}\beta_j \right\}^2 + \underbrace{\lambda \sum_{j=1}^{p} \beta_j^2}_{\substack{L2 \\ regularization}} \tag{8}$$

The ridge penalty factor is set to ensure that the coefficients of the estimated ridge regression stabilize to avoid overfitting. The closer lambda is to zero, the smaller the biasing constant, and the closer the ridge estimates will be to OLS. Standard errors are generally not available with penalized estimation methods because it is difficult to obtain a precise estimate of the bias which forms a major component of the mean square error. Reliable estimates of the bias are only available if unbiased estimates are available. Bootstrapped standard errors are however available in standard software packages such as Stata.

**Causality test.** To examine causality, Toda and Yamamoto's [43] Granger non-causality test is applied. The advantage of this method is that we can examine causality among variables of a different order of integration, and the method fits well with the ARDL procedure as the part of the information such as lag-length and maximum order of integration is used in the analysis. The maximum lag length is calculated as the sum of the maximum order of integration based on the unit root tests (*dmax*), and the maximum lag length (*l*) in the ARDL estimation. In the bivariate case, Toda and Yamamoto's test VAR is specified as follows:

$$y_t = \alpha_0 + \sum_{i=1}^{k} \alpha_{1i} y_{t-i} + \sum_{i=1}^{k} \alpha_{2i} x_{t-i} + dmax\ terms + u_{1,t} \tag{9}$$

$$x_t = \beta_0 + \sum_{i=1}^{k} \beta_{1i} y_{t-i} + \sum_{i=1}^{k} \beta_{2i} x_{t-i} + dmax\ terms + u_{2,t} \tag{10}$$

where $y$ and $x$ are time series variables, and the dmax terms are the additional 1 lag of the endogenous variables which are treated as exogenous to correct for model stability [43]. Hence, in Eq (9), Granger causality from $x$ to $y$ implies that $\alpha_{2i} \neq 0 \forall i$. Similarly, in Eq (10), Granger causality from $y$ to $x$ implies that $\beta_{2i} \neq 0 \forall i$. Dynamic stability of the test VAR model and robust causality outcomes are derived when the inverse roots of the autoregressive characteristic polynomial lie within the positive and negative unity. Where the inverse roots lie outside the unit circle, this can be corrected by including appropriate dmax terms, a trend term, or structural break dummies as exogenous instruments in the test VAR system.

## Results and discussion

### Basic statistics and index development

As noted from Tables 1 and 2, there is a strong positive and significant correlation between tourist arrivals, tourism receipts, official exchange rates, and real GDP per worker. Although correlation does not imply cointegration, the strength of the correlation and level of significance can influence the statistical significance of the long-run association.

Table 3 below summarizes the PCA results. Noting the vast differences in the mean value of arrivals and receipts, the PCA is run on the log of both indicators. The eigenvalue for arrivals exceeds 1 which indicates its relevance in the index [11]. The factor loading of the first component reveals that arrivals and receipts enter the first component with a similar weight [11]. Around 88 percent of the variation in the tourism index is explained by tourist arrivals.

### Unit root, structural breaks, and cointegration

Table 4 presents the results of the unit root tests. The results indicate that the variables are stationary in their first differences and suitable for the subsequent estimations. The optimal value of $k^*$ is determined by minimizing the sum of squared residuals obtained from Eq (5).

Table 5 presents the results of the multiple break test. Both the identified breaks were negative and significant in the subsequent estimations. The year 2007 could reflect the Nuku'alofa riots and the subsequent declaration of a state of emergency. The second break, 2010, could reflect the lagged effect of the Tonga Tsunami and earthquake in late 2009 [44].

Table 6 presents the results of the Bounds test for cointegration. Cointegration is strongly suggested at the 1 percent level in the estimated models.

**Table 1. Descriptive statistics.**

| Statistics | Real GDP per worker | Capital stock per worker | Tourist arrivals | Tourism receipts | Official exchange rate (TOP/USD) | Real exchange rate (TOP/USD) |
|---|---|---|---|---|---|---|
| Mean | 5875.87 | 18071.11 | 42342 | 25408068 | 1.87 | 2.31 |
| Median | 5850.33 | 17261.20 | 43850 | 20416167 | 1.94 | 2.17 |
| Maximum | 6864.91 | 24742.23 | 62500 | 54628384 | 2.27 | 3.35 |
| Minimum | 5059.86 | 12708.26 | 26000 | 11506932 | 1.23 | 1.70 |
| Std. Dev. | 532.84 | 3775.08 | 10323 | 12670503 | 0.31 | 0.43 |
| Skewness | 0.22 | 0.38 | 0.01 | 0.89 | -0.73 | 0.95 |
| Kurtosis | 2.29 | 1.86 | 2.14 | 2.52 | 2.59 | 3.29 |
| Jarque-Bera | 0.70 | 1.87 | 0.74 | 3.41 | 2.28 | 3.75 |
| Probability | 0.70 | 0.39 | 0.69 | 0.18 | 0.32 | 0.15 |

Estimated in EViews 10.

**Table 2. Correlation matrix.**

| Statistics | Real GDP per worker | Capital stock per worker | Tourist arrivals | Tourism receipts | Official exchange rate (TOP/USD) | Real exchange rate (TOP/USD) |
|---|---|---|---|---|---|---|
| Real GDP per worker | 1.00 | | | | | |
| Capital stock per worker | 0.95[A] | 1.00 | | | | |
| Tourist arrivals | 0.91[A] | 0.93[A] | 1.00 | | | |
| Tourism receipts | 0.84[A] | 0.91[A] | 0.80[A] | 1.00 | | |
| Official exchange rate (TOP/USD) | 0.77[A] | 0.62[A] | 0.72[A] | 0.41[B] | 1.00 | |
| Real exchange rate (TOP/USD) | -0.24 | -0.47[B] | -0.40[B] | -0.51[B] | 0.29 | 1.00 |

Estimated in EViews 10. A, B indicates significance at the 1 and 5 percent levels.

## Long-run results

Tables 7–9 present the results of the growth model discussed earlier. A total of 6 models are considered. The lag length for each variable in the ARDL model is determined by the Schwarz information criteria. The arrivals models (Table 7) generally indicate that tourism promotes growth. The exchange rates, both real and nominal, are appropriately signed which suggests that an increase of the Tongan Pa'anga against the USD reduces growth. Structural breaks have negative effects on growth, although only the break for 2007 is significant across the estimates irrespective of the tourism or exchange rate assumption.

The receipts model returns mixed effects on tourism and growth. Considering the real exchange rate, tourism receipts promote growth when estimated using the ARDL or DOLS methods (Table 8). Considering the nominal exchange rate, the FMOLS estimates also suggest this outcome. However, the effect is negative according to the CCR estimator irrespective of whether the model is estimated using annual or quarterly data (Table 8).

The most consistent results are obtained with the PCA tourism development index models (Table 9). The growth effect of tourism is between 0.02 to 0.04, respectively. The results with the tourism indicator (Table 9) are qualitatively similar to those with tourism arrivals (Table 7). The predominance of tourism arrivals in the index which explains 88 percent of the variations in the index could explain this similarity. Unlike earlier results (Tables 7 and 8), the effects are consistent across methods, data frequencies, and exchange rate assumptions. Therefore, developing an overall tourism index is better suited to assess the tourism-growth association. The capital share is between 0.27 to 0.37 which is smaller than those from the earlier estimates (Tables 7 and 8). Consistency is also evident in the exchange rates despite the unit of measurement and is in between 0.08 to 0.12. Overall, the growth effect of tourism is small, positive, and statistically significant in Tonga.

The results indicate that tourism is an important driver of long-run growth in Tonga like other Pacific islands [45, 46]. However, the effect of tourism is noticeably smaller than in earlier studies. This suggests that although the tourism sector does affect growth, it requires

**Table 3. PCA results for weighted tourism development index.**

| Component | Eigenvalue | Proportion | Factor Loadings | Correlation with Index |
|---|---|---|---|---|
| Tourist arrivals | 1.75 | 0.88 | 0.71 | 0.99 |
| Tourism receipts | 0.25 | 0.12 | 0.71 | 0.75 |

Estimated in Stata 16 and EViews 10.

**Table 4. Unit root tests.**

| Variables | Augmented Dickey-Fuller | | Phillips-Perron | | Flexible Fourier | |
|---|---|---|---|---|---|---|
| | Level | 1st Difference | Level | 1st Difference | Level | 1st Difference |
| $\ln y_t$ | -0.3881(0) | -3.2187(0)[A] | -0.5078(1) | -3.2348(1)[A] | 0.4181(1) | 9.9236(1)[A] |
| $\ln k_t$ | -0.5878(2) | -2.9391(1)[A] | -0.1502(1) | -2.8730(0)[A] | 0.5418(1) | 5.4164(1)[B] |
| $\ln ta_t$ | -1.0098(0) | -4.6883(0)[A] | -0.8684(7) | -5.4372(14)[A] | 1.2051(1) | 11.3438(1)[A] |
| $\ln tr_t$ | -1.0030(0) | -6.1533(0)[A] | -0.8393(1) | -6.1533(0)[A] | 2.3456(0) | 23.9238(1)[A] |
| $\ln ti_t$ | -1.0099(0) | -4.6885(0)[A] | -0.8684(7) | -5.4372(14)[A] | 1.1052(0) | 11.3344(1)[A] |
| $\ln ner_t$ | -2.4373(3) | -2.7830(1)[A] | -1.8291(1) | -2.8793(1)[A] | 1.0209(2) | 9.5244(1)[A] |
| $\ln rer_t$ | -1.9558(1) | -2.6434(0)[A] | -1.4472(2) | -2.6434(0)[A] | 1.7497(2) | 6.7695(1)[A] |

Estimated in EViews 10 and Microsoft Excel 2016. Lag used in ADF, FF, and Bandwidth in PP are indicated in (.). A, B, C indicates stationarity at 1,5,10 percent. Test conducted with intercept only. The test is based on annual series.

further development. For this reason, factors promoting tourism demand are necessary, and building resilience is critical. Policymakers in Tonga could capitalize on factors such as a favorable word of mouth by establishing good reviews on sites such as TripAdvisor and distinguishing the Tongan tourism experience from the other PICs. Promoting a safe and secure environment for tourists by strengthening law enforcement, and knowledge of overall tourism demand elasticities may prove beneficial.

The estimated models (Tables 7–9) are free from auto-correlated residuals and heteroscedasticity, and the residuals are normally distributed. The estimates do not exhibit the endogeneity bias, have correct functional forms, and are stable. These results are not presented to conserve space but are available upon request.

## Multicollinearity test and ridge estimates

Noting the consistency of the results from the PCA models (Table 9), the PCA model is re-visited using the ridge regression technique. Strong multicollinearity between the explanatory variables is found evidenced by the strong correlation between the variables (Table 2) and according to the Farrar-Glauber test (Table 10). Minor differences are found between the ridge and cointegrating models (Tables 7–10). The ridge coefficient evolution plot indicates that the estimated coefficients stabilize at the chosen ridge lambda penalty factor (Fig 2).

## Causality test

To undertake causality analysis, we rely on the PCA tourism index-real exchange rate model. We set a lag of 1 in the test VAR model which is within the sum of the order of integration and maximum lag of the ARDL model, both which are 1, respectively [43; 45]. The significant causal relations are reported in Table 11 below. Notably, the causality results in Table 11

**Table 5. Multiple break test.**

| Hypothesized number of breaks | Scaled F-Test Statistic | Critical Value |
|---|---|---|
| 0 vs. 1 | 18.65[A] | 8.58 |
| 1 vs. 2 | 10.61[A] | 10.13 |
| 2 vs. 3 | 4.40 | 11.14 |

Break dates: 2007; 2010

Estimated in EViews 10. A indicates break at the 5 percent level. Break variables: Intercept.

**Table 6. Bounds test.**

| Model | Break Adjusted F-Test Statistic | |
|---|---|---|
| **Tourist arrivals** | Annual | Quarterly |
| Real exchange rate | 32.29[A] | 22.60[A] |
| Nominal exchange rate | 12.12[A] | 6.80[A] |
| **Tourism receipts** | | |
| Real exchange rate | 15.18[A] | 11.08[A] |
| Nominal exchange rate | 17.24[A] | 6.03[A] |
| **PCA tourism index** | | |
| Real exchange rate | 11.46[A] | 22.58[A] |
| Nominal exchange rate | 6.48[A] | 8.83[A] |

Critical values at 1 percent: I(0) = 4.29; I(1) = 5.61. Model assumes intercept only.

Estimated in EViews 10. A indicates cointegration at the 1 percent level.

indicate that tourism, real exchange rates, and capital granger cause growth. Table 11 further indicates that the real exchange rate granger causes tourism and that capital investments

**Table 7. Tourist arrivals models.**

| Variables | Annual | | | | | Quarterly | | | | |
|---|---|---|---|---|---|---|---|---|---|---|
| | ARDL | FMOLS | CCR | DOLS | MEAN | ARDL | FMOLS | CCR | DOLS | MEAN |
| Panel A: arrivals and real exchange rates | | | | | | | | | | |
| $\ln k_t$ | 0.42[A] | 0.32[A] | 0.25[A] | 0.31[A] | 0.33[B] | 0.41[A] | 0.30[A] | 0.30[A] | 0.33[A] | 0.34[A] |
| | [0.0480] | [0.0581] | [0.0794] | [0.0613] | [0.0705] | [0.0309] | [0.0378] | [0.0357] | [0.0466] | [0.0466] |
| $\ln ta_t$ | 0.09[B] | 0.14[A] | 0.16[A] | 0.16[A] | 0.14[B] | 0.09[A] | 0.14[A] | 0.14[A] | 0.15[A] | 0.13[B] |
| | [0.0319] | [0.0349] | [0.0524] | [0.0382] | [0.0330] | [0.0179] | [0.0231] | [0.0225] | [0.0290] | [0.0271] |
| $\ln rer_t$ | 0.12[A] | 0.09[A] | 0.10[A] | 0.08[A] | 0.10[A] | 0.11[A] | 0.10[A] | 0.10[A] | 0.08[A] | 0.10[A] |
| | [0.0124] | [0.0216] | [0.0285] | [0.0245] | [0.0171] | [0.0110] | [0.0150] | [0.0150] | [0.0193] | [0.0193] |
| D1 | -0.05[A] | -0.06[A] | -0.05[A] | -0.07[A] | -0.06[A] | -0.05[A] | -0.06[A] | -0.06[A] | -0.07[A] | -0.06[A] |
| | [0.0061] | [0.0114] | [0.0165] | [0.0132] | [0.0096] | [0.0055] | [0.0081] | [0.0084] | [0.014] | [0.0082] |
| D2 | -0.03[A] | -0.01 | 0.01 | -0.02 | 0.01 | -0.03[A] | -0.01 | -0.01 | -0.02 | 0.01 |
| | [0.0099] | [0.0165] | [0.0221] | [0.0191] | [0.0222] | [0.0096] | [0.0117] | [0.0116] | [0.0150] | [0.0096] |
| Intercept | 3.43[A] | 3.97[A] | 4.41[A] | 3.88[A] | 3.93[A] | 3.59[A] | 4.17[A] | 4.16[A] | 3.86[A] | 3.95[A] |
| | [0.1991] | [0.1991] | [0.0165] | [0.1991] | [0.4018] | [0.1773] | [0.2136] | [0.2045] | [0.2672] | [0.2672] |
| Panel B: arrivals and nominal exchange rates | | | | | | | | | | |
| $\ln k_t$ | 0.37[A] | 0.28[A] | 0.22[B] | 0.27[A] | 0.29[A] | 0.34[A] | 0.23[A] | 0.23[A] | 0.27[A] | 0.27[B] |
| | [0.0522] | [0.0628] | [0.0964] | [0.0618] | [0.0618] | [0.0584] | [0.0421] | [0.0404] | [0.0469] | [0.0519] |
| $\ln ta_t$ | 0.08[B] | 0.08[B] | 0.12[C] | 0.11[B] | 0.10[B] | 0.08[A] | 0.10[A] | 0.11[A] | 0.11[A] | 0.10[A] |
| | [0.0225] | [0.0417] | [0.0601] | [0.0406] | [0.0206] | [0.0297] | [0.0271] | [0.0265] | [0.0303] | [0.0141] |
| $\ln ner_t$ | 0.10[A] | 0.12[A] | 0.12[A] | 0.11[A] | 0.11[A] | 0.10[A] | 0.13[A] | 0.13[A] | 0.11[A] | 0.11[A] |
| | [0.0174] | [0.0299] | [0.0343] | [0.0302] | [0.0082] | [0.0234] | [0.0202] | [0.0192] | [0.0233] | [0.0126] |
| D1 | -0.07[A] | -0.07[A] | -0.06[A] | -0.07[A] | -0.07[A] | -0.07[A] | -0.07[A] | -0.07[A] | -0.07[A] | -0.07[A] |
| | [0.0069] | [0.0113] | [0.0136] | [0.0116] | [0.0050] | [0.0087] | [0.0079] | [0.0079] | [0.0091] | [0.0091] |
| D2 | -0.05[A] | -0.01 | -0.01 | -0.02 | -0.02 | -0.04[A] | -0.01 | -0.01 | -0.02 | -0.02 |
| | [0.0166] | [0.0177] | [0.0226] | [0.0185] | [0.0189] | [0.0176] | [0.0125] | [0.0120] | [0.0143] | [0.0143] |
| Intercept | 4.12[A] | 4.93[A] | 5.21[A] | 4.85[A] | 4.78[A] | 4.47[A] | 5.23[A] | 5.19[A] | 4.84[A] | 4.84[A] |
| | [0.3907] | [0.4745] | [0.0613] | [0.4955] | [0.4647] | [0.4587] | [0.3349] | [0.3186] | [0.3836] | [0.3836] |

Estimated in EViews 10 and Microsoft Excel. A, B, C indicates statistical significance at the 1, 5, and 10 percent levels. Standard errors in [.].

**Table 8. Tourist receipts models.**

| Variables | Annual | | | | | Quarterly | | | | |
|---|---|---|---|---|---|---|---|---|---|---|
| | ARDL | FMOLS | CCR | DOLS | MEAN | ARDL | FMOLS | CCR | DOLS | MEAN |
| | Panel A: receipts and real exchange rates | | | | | | | | | |
| ln $k_t$ | 0.51[A] | 0.56[A] | 0.58[B] | 0.51[A] | 0.54[A] | 0.50[A] | 0.53[A] | 0.53[B] | 0.55[A] | 0.53[A] |
| | [0.0226] | [0.0585] | [0.0295] | [0.0618] | [0.0356] | [0.0397] | [0.0394] | [0.0388] | [0.0559] | [0.0206] |
| ln $tr_t$ | 0.01[C] | -0.01 | -0.01[C] | 0.07[B] | 0.02 | 0.01[B] | -0.01 | -0.01[C] | -0.01 | -0.01 |
| | [0.0059] | [0.0161] | [0.0081] | [0.0121] | [0.0379] | [0.0065] | [0.0111] | [0.0107] | [0.0121] | [0.0010] |
| ln $rer_t$ | 0.15[A] | 0.10[A] | 0.06[A] | 0.18[A] | 0.12 | 0.14[A] | 0.10[A] | 0.10[A] | 0.08[A] | 0.11[B] |
| | [0.0144] | [0.0323] | [0.0179] | [0.0099] | [0.0532] | [0.0257] | [0.0221] | [0.0216] | [0.0318] | [0.0252] |
| D1 | -0.03[A] | -0.05[A] | -0.05[A] | -0.04[A] | -0.04[B] | -0.03[A] | -0.04[A] | -0.04[A] | -0.05[A] | -0.04[B] |
| | [0.0051] | [0.0146] | [0.0084] | [0.0029] | [0.0096] | [0.0156] | [0.0102] | [0.0105] | [0.0146] | [0.0082] |
| D2 | -0.03[A] | -0.03 | -0.05[A] | -0.07[A] | -0.05 | -0.03[A] | -0.02 | -0.02 | -0.04[A] | -0.03[C] |
| | [0.0104] | [0.0177] | [0.024] | [0.0088] | [0.0191] | [0.0211] | [0.0156] | [0.0155] | [0.0223] | [0.0096] |
| Intercept | 3.34[A] | 3.44[A] | 3.28[A] | 2.38[A] | 3.11[A] | 3.42[A] | 3.68[A] | 3.69[A] | 3.47[A] | 3.57[A] |
| | [0.1695] | [0.4050] | [0.1871] | [0.1789] | [0.4911] | [0.3257] | [0.2714] | [0.2664] | [0.3828] | [0.1401] |
| | Panel B: receipts and nominal exchange rates | | | | | | | | | |
| ln $k_t$ | 0.28[A] | 0.24[A] | 0.35[A] | 0.19[A] | 0.27[B] | 0.27[A] | 0.18[A] | 0.19[A] | 0.24[A] | 0.22[B] |
| | [0.0226] | [0.0825] | [0.0827] | [0.0699] | [0.0676] | [0.0861] | [0.0698] | [0.0647] | [0.0860] | [0.0424] |
| ln $tr_t$ | 0.04[A] | 0.03[C] | 0.01 | 0.09[B] | 0.04 | 0.03[A] | 0.03[B] | 0.03[A] | 0.03[C] | 0.03[A] |
| | [0.0059] | [0.0151] | [0.0146] | [0.0255] | [0.0340] | [0.0093] | [0.0130] | [0.0121] | [0.0161] | [0.0005] |
| ln $ner_t$ | 0.20[A] | 0.20[A] | 0.14[A] | 0.25[A] | 0.20[B] | 0.18[A] | 0.22[A] | 0.22[A] | 0.19[A] | 0.20[A] |
| | [0.0258] | [0.0323] | [0.0351] | [0.0365] | [0.0450] | [0.0324] | [0.0298] | [0.0275] | [0.0371] | [0.0206] |
| D1 | -0.05[A] | -0.05[A] | -0.06[A] | -0.05[A] | -0.05[A] | -0.05[A] | -0.04[A] | -0.04[A] | -0.05[A] | -0.05[A] |
| | [0.0057] | [0.0091] | [0.0088] | [0.0060] | [0.0050] | [0.0161] | [0.0079] | [0.0088] | [0.0099] | [0.0058] |
| D2 | -0.03[B] | 0.01 | -0.02 | -0.04[B] | -0.02 | -0.02[A] | 0.02 | 0.02 | 0.01 | 0.01 |
| | [0.0133] | [0.0190] | [0.0183] | [0.0203] | [0.0216] | [0.0269] | [0.0162] | [0.0154] | [0.0199] | [0.0189] |
| Intercept | 5.08[A] | 5.77[A] | 4.92[A] | 5.19[A] | 5.24[A] | 5.28[A] | 6.22[A] | 6.16[A] | 5.79[A] | 5.86[A] |
| | [0.4035] | [0.5765] | [0.5666] | [0.5000] | [0.3703] | [0.7177] | [0.4882] | [0.4539] | [0.5992] | [0.4324] |

Estimated in EViews 10 and Microsoft Excel. A, B, C indicates statistical significance at the 1, 5, and 10 percent levels. Standard errors in [.].

granger cause the real exchange rate and tourism, respectively which reaffirms the findings in Table 9. In this regard, predicting Tonga's economic growth requires a careful analysis of the effect of tourism, real exchange rate, and capital noting the potential inter-relationships. Fig 3 suggests that the test VAR model is stable and hence causality outcomes are reliable.

## Further tests

Two further tests are conducted with the PCA models. First, nonlinear effects of tourism are considered. On nonlinearity, we draw insights from Balsalobre-Lorente et al. [47] and invoke the partial sum decomposition technique to decompose positive and negative changes in tourism. Then, the threshold effects of tourism on growth is considered through threshold regressions [48], and by including the square of tourism in the specification [49]. Nonlinear effects are rejected in all specifications. Second, the model is re-estimated using Bayesian techniques [50]. The posterior mean of the Bayesian estimator resembles the estimates in Tables 7–10. These results are not presented to conserve space but are available upon request.

**Table 9. PCA tourism development index models.**

| Variables | Annual | | | | | Quarterly | | | | |
|---|---|---|---|---|---|---|---|---|---|---|
| | ARDL | FMOLS | CCR | DOLS | MEAN | ARDL | FMOLS | CCR | DOLS | MEAN |
| Panel A: PCA index and real exchange rates | | | | | | | | | | |
| ln $k_t$ | 0.32[A] | 0.33[A] | 0.31[A] | 0.32[A] | 0.32[A] | 0.37[A] | 0.30[A] | 0.30[A] | 0.33[A] | 0.33[A] |
| | [0.0717] | [0.0589] | [0.0274] | [0.0536] | [0.0081] | [0.0361] | [0.0378] | [0.0357] | [0.0465] | [0.0332] |
| ln $ti_t$ | 0.04[A] | 0.04[A] | 0.04[A] | 0.04[A] | 0.04[A] | 0.03[A] | 0.04[A] | 0.04[A] | 0.04[A] | 0.04[A] |
| | [0.0099] | [0.0091] | [0.0042] | [0.0086] | [0.0004] | [0.0053] | [0.0059] | [0.0058] | [0.0074] | [0.0050] |
| ln $rer_t$ | 0.10[A] | 0.10[A] | 0.09[A] | 0.08[A] | 0.09[A] | 0.13[A] | 0.10[A] | 0.10[A] | 0.08[A] | 0.10[B] |
| | [0.0144] | [0.0589] | [0.0089] | [0.0099] | [0.0096] | [0.0134] | [0.0589] | [0.0151] | [0.0193] | [0.0206] |
| D1 | -0.07[A] | -0.06[A] | -0.05[A] | -0.07[A] | -0.07[A] | -0.05[A] | -0.06[A] | -0.06[A] | -0.07[A] | -0.07[A] |
| | [0.0122] | [0.0115] | [0.0046] | [0.0115] | [0.0050] | [0.0064] | [0.0081] | [0.0046] | [0.0104] | [0.0050] |
| D2 | -0.01 | -0.02 | -0.01 | -0.02 | -0.02[C] | -0.01 | -0.01 | -0.01 | -0.02 | -0.02[C] |
| | [0.0022] | [0.0167] | [0.0074] | [0.0167] | [0.0058] | [0.0115] | [0.0117] | [0.0116] | [0.0150] | [0.0058] |
| Intercept | 5.52[A] | 5.40[A] | 5.53[A] | 5.53[A] | 5.50[A] | 4.91[A] | 5.69[A] | 5.65[A] | 5.43[A] | 5.50[A] |
| | [0.6906] | [0.5735] | [0.2647] | [0.0167] | [0.0636] | [0.3491] | [0.3659] | [0.3464] | [0.4503] | [0.0636] |
| Panel B: PCA index and nominal exchange rates | | | | | | | | | | |
| ln $k_t$ | 0.28[A] | 0.31[A] | 0.29[A] | 0.27[A] | 0.29[A] | 0.31[A] | 0.29[A] | 0.28[A] | 0.27[A] | 0.29[A] |
| | [0.0675] | [0.0631] | [0.0334] | [0.0541] | [0.0171] | [0.0636] | [0.0427] | [0.0362] | [0.0470] | [0.0171] |
| ln $ti_t$ | 0.02[A] | 0.02[A] | 0.02[A] | 0.03[A] | 0.02[B] | 0.03[A] | 0.03[A] | 0.02[A] | 0.03[A] | 0.03[B] |
| | [0.0097] | [0.0107] | [0.0061] | [0.0091] | [0.0050] | [0.0090] | [0.0070] | [0.0065] | [0.0077] | [0.0050] |
| ln $ner_t$ | 0.12[A] | 0.12[A] | 0.11[A] | 0.11[A] | 0.12[A] | 0.10[A] | 0.12[A] | 0.12[A] | 0.11[A] | 0.11[A] |
| | [0.0303] | [0.0299] | [0.0154] | [0.0264] | [0.0058] | [0.0226] | [0.0205] | [0.0190] | [0.0233] | [0.0096] |
| D1 | -0.07[A] | -0.07[A] | -0.07[A] | -0.07[A] | -0.07[A] | -0.07[A] | -0.07[A] | -0.07[A] | -0.07[A] | -0.07[A] |
| | [0.0102] | [0.0113] | [0.0055] | [0.0101] | [0.0005] | [0.0083] | [0.0080] | [0.0077] | [0.0091] | [0.0005] |
| D2 | -0.02 | -0.02 | -0.02 | -0.02 | -0.02[C] | -0.04 | -0.02[B] | -0.02[C] | -0.02 | -0.02[C] |
| | [0.0195] | [0.0177] | [0.0087] | [0.0161] | [0.0005] | [0.0173] | [0.0079] | [0.0111] | [0.0144] | [0.0005] |
| Intercept | 5.85[A] | 5.58[A] | 5.83[A] | 6.02[A] | 5.82[A] | 5.61[A] | 5.78[A] | 5.89[A] | 5.95[A] | 5.81[A] |
| | [0.6481] | [0.6091] | [0.3219] | [0.5187] | [0.1813] | [0.6127] | [0.4107] | [0.3476] | [0.4503] | [0.1493] |

Estimated in EViews 10 and Microsoft Excel. A, B, C indicates statistical significance at the 1, 5, and 10 percent levels. Standard errors in [.].

## Conclusions

In this study, data and method triangulation are proposed as potential solutions to the publication bias problem [3–5]. The specification incorporates the effects of exchange rates, capital-labor ratio, and structural breaks. On method triangulation, the autoregressive distributed lag,

**Table 10. PCA ridge estimates.**

| Variable | Coefficient | Standard Error | t-Stat | P-value |
|---|---|---|---|---|
| ln $k_t$ | 0.3077[A] | 0.0584 | 5.2700 | <0.01 |
| ln $ti_t$ | 0.0403[A] | 0.0094 | 4.3000 | <0.01 |
| ln $rer_t$ | 0.0808[A] | 0.0234 | 3.4600 | <0.01 |
| D1 | -0.0682[A] | 0.0125 | -5.4500 | <0.01 |
| D2 | -0.0201 | 0.0181 | -1.1100 | 0.28 |
| Intercept | 5.6146[A] | 0.5645 | 9.9400 | <0.01 |

Farrar-Glauber Multicollinearity Test: $\chi^2$ = 111.5685, P-value: <0.01; Mean VIF = 11.0959

Estimated in EViews 10 and Stata 16. A-significant at 1 percent.

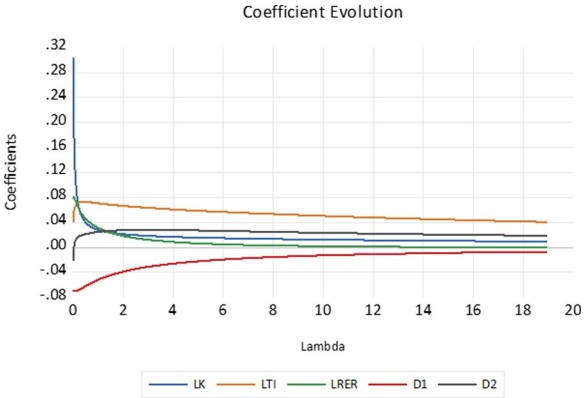

**Fig 2. Ridge trace evolution plot.** Estimated in EViews 10.

fully modified least squares, dynamic least squares, canonical cointegrating regression, and ridge regression are considered. On data triangulation, tourism is measured by arrivals, receipts, and an index of tourism performance developed using principal component analysis. A data frequency conversion method is used to derive the quarterly data series. Tonga is used as a case study over the sample period 1995 to 2018. Tonga is chosen given the importance of tourism and the dearth of country-specific research for this country.

The findings indicate that the size of the effect of tourism in growth regressions depends on the measure of tourism. Consistent results across all three dimensions are obtained using the PCA index models. Tourism has small, positive, and significant growth effects. A 1 percent increase in tourism would increase growth by about 0.02 to 0.04 percent, ceteris paribus. A 1 percent decline of the Tongan Pa'anga against the USD would increase growth by about 0.10 to 0.12 percent, ceteris paribus. The capital elasticity has an average value of about 0.32. The structural break which represents the 2006/07 Nuku'alofa riots has significant negative effects. Unidirectional causality from tourism to growth is found with the causality test.

Nonetheless, the study could have benefitted from a larger sample size but was restricted to a sample of 1995 to 2018 due to a lack of earlier tourism data. Yet, the key scientific implication/contribution is the development of a cohesive framework that attempts to solve publication biases in the tourism-growth literature. Synthesizing the literature, the framework developed draws from the Solow-Swan growth model, controls for exchange rates and structural breaks, and utilizes an overall indicator of tourism performance. Multiple methods which correct for small sample and endogeneity biases, and multicollinearity are used. Nonlinearity is also considered but found statistically insignificant. Future research can apply the framework advanced in this study to potentially circumvent the publication bias critique.

**Table 11. Causality results.**

| Causality Direction | Chi-Squared Statistic |
|---|---|
| $\ln k_t \rightarrow \ln y_t$ | 11.14[A] |
| $\ln ti_t \rightarrow \ln y_t$ | 5.89[B] |
| $\ln rer_t \rightarrow \ln y_t$ | 27.42[A] |
| $\ln k_t \rightarrow \ln ti_t$ | 3.35[C] |
| $\ln rer_t \rightarrow \ln ti_t$ | 11.01[A] |
| $\ln k_t \rightarrow \ln rer_t$ | 5.96[B] |

Estimated in EViews 10. A, B, and C indicate causality at 1, 5, and 10 percent levels.

## Inverse Roots of AR Characteristic Polynomial

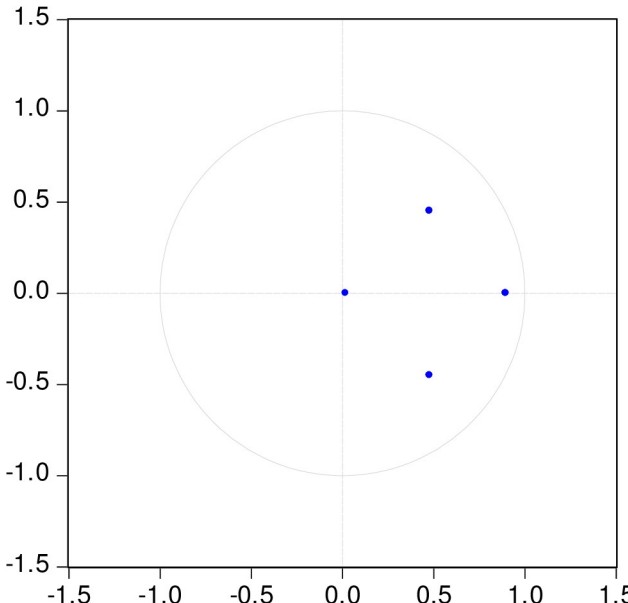

**Fig 3. Inverse roots characteristic stability plot.** Estimated in EViews 10. Inverse roots within the unit circle indicate stable estimates.

Based purely on the results, any policy promoting tourism in Tonga would contribute to economic growth of the country. The practical policy implications need to consider the positive and significant growth effect of tourism, and negative effects arising from political issues and other exogenous shocks. Based on the authors knowledge of Tonga and its tourism industry, policymakers need to make careful decisions in how capital projects are implemented, and how budget shares are allocated to an industry like tourism. This is because resources are limited and there are many other equally urgent competing social projects. To develop the tourism sector, enabling investments in basic infrastructures such as roads, airports and international and domestic air transportation, information and communication technology, public amenities, and easing of restrictions to access financial services is needed. Demand-side factors such as the sensitivity of tourism demand to price and income shocks, and a favourable word of mouth is also important. Tailor-made tourist packages catered for Australian and New Zealand tourists, and the establishment of direct travel routes may also prove beneficial.

However, policy decisions to invest in the tourism industry and related areas need to be cautioned based on the noticeably small growth effects found in this study. This implies that although the tourism sector influences growth, its magnitude is small relative to competing destinations, and requires further development. Additionally, the effect of COVID 19 on the tourism industry is unprecedented and requires a radical shift in the way countries depend on tourism [51]. Given the small positive impact of tourism on growth in Tonga, alternative growth strategies such as agriculture, back-office data processing, and call centers that work in tandem with tourism are needed. Further, Tonga needs to pay attention to political stability to avoid the negative effects of tourism on growth. Further research is thus needed to address how much tourism contributes multiplicatively to other industries, like agriculture, and the level of direct and induced employment generated through tourism activities.

## Supporting information

**S1 Data.**
(XLSX)

## Acknowledgments

The authors thank the academic editor and anonymous reviewers for their useful comments which have considerably improved this paper. The authors also thank Associate Professor Saten Kumar, Dr Antony Andrews, and Mr. Sean Kimpton of AUT for insights on Ridge and Bayesian analysis, and for proof-reading the revised text. Nikeel would also like to thank Nandani for her constant encouragement and for proofreading the revised text.

## Author Contributions

**Conceptualization:** Nikeel Nishkar Kumar, Arvind Patel.

**Data curation:** Nikeel Nishkar Kumar, Ravinay Amit Chandra, Navneet Nimesh Kumar.

**Formal analysis:** Nikeel Nishkar Kumar.

**Investigation:** Nikeel Nishkar Kumar, Arvind Patel.

**Methodology:** Nikeel Nishkar Kumar, Arvind Patel.

**Project administration:** Nikeel Nishkar Kumar, Arvind Patel, Ravinay Amit Chandra.

**Resources:** Arvind Patel, Ravinay Amit Chandra, Navneet Nimesh Kumar.

**Software:** Nikeel Nishkar Kumar, Navneet Nimesh Kumar.

**Supervision:** Arvind Patel.

**Validation:** Nikeel Nishkar Kumar, Arvind Patel.

**Writing – original draft:** Nikeel Nishkar Kumar, Ravinay Amit Chandra, Navneet Nimesh Kumar.

**Writing – review & editing:** Nikeel Nishkar Kumar, Arvind Patel, Ravinay Amit Chandra, Navneet Nimesh Kumar.

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
