## [Decision Letter · Decision Letter 0]

20 May 2021

PONE-D-21-05441

Publication Bias and the Tourism Led Growth Hypothesis

PLOS ONE

Dear Dr. Kumar,

Thank you for submitting your manuscript to PLOS ONE. After careful consideration, we feel that it has merit but does not fully meet PLOS ONE’s publication criteria as it currently stands. Therefore, we invite you to submit a revised version of the manuscript that addresses the points raised during the review process.

Please address the comments and questions from the reviewers. I have a few additional comments and suggestions as listed below.

We look forward to receiving your revised manuscript.

Kind regards,

Hiranya K. Nath, Ph.D.

Academic Editor

PLOS ONE

Journal Requirements:

2. Please include a copy of Table xxxx which you refer to in your text on page 13.

Additional Editor Comments :

The authors should address the reviewers' comments and questions. I would like to add a few comments and questions of mine:

1) The sample period includes only 24 years of data. The authors should discuss the potential issues associated with the small size of the sample, particularly in the context of the methods used in the paper. Although they comment at the end, they should discuss it much earlier in the text.

2) The authors should be careful in discussing the policy implications. Purely based on the results, they can say that any policy promoting tourism in Tonga would contribute to economic growth of the country. The specific policy prescriptions that they have included do not necessarily follow from the data analysis presented in the paper. They are based on the authors' knowledge of the country and its tourism industry. They should clearly acknowledge that.

3) Although the authors motivate the paper by referring to the potential impacts of Covid-19 on the tourism industry in Tonga, I am afraid there is not much in terms of specific policy intervention in such a situation. First, the dataset does not cover the pandemic period. Second, although the data analysis presents evidence of negative impacts of two earlier events in 2007 and 2010 on growth, it is not clear if they work through the tourism channel. Further, those were destination specific events while Covid-19 has impacted both sources and destination for the tourism industry.

4) The null hypotheses stated on p. 9 of the paper are not correct. The null hypothesis of an Augmented Dickey-Fuller test is that the underlying series has a unit root and the rejection of the null hypothesis indicates that the series is stationary. The statement for the BEL test seems to be inconsistent as well.

5) The results with the tourism indicator (Table 10) seems to be qualitatively similar to those with tourism arrivals (Table 8). The predominance of tourism arrivals in the index (88% of the variations in the index being driven by tourism arrivals) could explain this similarity.

Reviewers' comments:

Reviewer's Responses to Questions

**Comments to the Author**

1. Is the manuscript technically sound, and do the data support the conclusions?

Reviewer #1: Yes

Reviewer #2: Partly

2. Has the statistical analysis been performed appropriately and rigorously? 

Reviewer #1: Yes

Reviewer #2: Yes

3. Have the authors made all data underlying the findings in their manuscript fully available?

Reviewer #1: Yes

Reviewer #2: Yes

4. Is the manuscript presented in an intelligible fashion and written in standard English?

Reviewer #1: Yes

Reviewer #2: Yes

5. Review Comments to the Author

Reviewer #1: 1. A review of literature discussing why the author considers the issue of publication bias is necessary. Particularly, the author may provide a summary of how there is a tendency to report favourable and significant results in the tourism-growth literature. This would help supply a conceptual framework to the paper.

2. The author has selected many methods such as ARDL, DOLS, FMOLS and CCR in the estimation. However, it is not clear why these methods were selected. The suitability of the methods and their likely advantages in the study requires an explanation.

3. In addition to tourism indicator (tourist arrival, tourist receipt and tourism development index), the author considers only capital stock per worker, exchange rates (nominal and real) and two dummy variables. However, it is unclear why the study is restricted to only these independent variables.

4. There is a high degree of correlation (presented in table 2) between capital stock and tourist arrival, capital stock and tourist receipt and between nominal exchange rate and tourist arrival. Therefore, it would not be sound to use these variables in the same regression model.

5. The criteria of lag selection in the causality analysis and unit root test also needs some clarity.

6. The weights used in the calculation of the Tourism Development Index using PCA could be mentioned.

7. Authors should also highlight the key contribution of this research over the existing literature.

Reviewer #2: I think the paper is interesting and contains a remarkable orientation to provide information to academics and practitioners. The issue is relevant for influencing policy and practice.

The paper contents follow a logical scheme. In my opinion, methodology is presented properly, and it is the most relevant contribution of this work. The paper’s objective is interesting.

In P.11, Introduction section, authors talk about the practical implication demands more appropriate policy decisions towards recovery and resilience due to pandemics such as COVID-19. So, in Conclusions section it is expected to argue about policy implications as has been said in Introduction section, but the only sentence can be found in P.25 is “The practical implication is on more appropriate policy decisions for recovery, growth, and resilience through tourism. Enabling investments in basic infrastructures such as roads, airports and international and domestic air transportation, information and communication technology, public amenities, and easing of restrictions to access financial services by tourists would be beneficial. Demand-side factors such as the sensitivity of tourism demand to price and income shocks and the development of a beneficial word of mouth effect need to be re-examined. Tailor-made tourist packages catered for Australian and New Zealand tourists may prove beneficial as would the establishment of direct travel routes.” More reflections are needed. Policy recommendations must be highlighted as well as important contributions derived from this work to support one of its main strengths properly.

Certainly, the results indicate that tourism is an important driver of long-run growth in Tonga, like other Pacific islands. However, the paper confirms the overall effect of tourism is smaller than in earlier studies. This may suggest that although the tourism sector influences growth, its magnitude is small relative to competing destinations, and requires further development.

Also, some concerns are related to References along the paper. Here there are some examples but please, revise all:

- P. 9, L. 54: The paper mentions reference “[5,6]” but authors corresponding reference 5 have not been mentioned before.

- P.9, L. 60: It happens the same with “[8]”

- P. 10, L. 76: The same with “[12]”

- P. 10, L. 80: The same with “[14]”

- P. 11, L. 98: The same with “[17]”

- P. 11, L. 100: The same with “[18, 19, 20]”

6. PLOS authors have the option to publish the peer review history of their article (what does this mean?). If published, this will include your full peer review and any attached files.

Reviewer #1: No

Reviewer #2: No

---

## [Author Response · Author response to Decision Letter 0]

5 Jul 2021

Response to Editors and Reviewers Comments (Please also see the attached document alongside the revised manuscript). The contents of the reviewer response document are reproduced here.

We would like to firstly thank the handling academic Editor of PLOS ONE, Professor Hiranya Nath, and the anonymous reviewers for their useful comments which have helped to considerably improve the quality of this paper. We would also like to thank the journal’s editorial team for efficiently moving the paper through the peer-review process. Given the comments received, we have re-worked key parts of the paper including the introduction, policy discussions, and references. We have also re-worked parts of the methodology to make the discussion on why particular methods are chosen clearer to the reader. Given the comments on multicollinearity, we have added a supporting section on ridge regression that attempts to remedy this shortcoming. The contributions of the study are also reworked in the introduction. The discussions on why the methods are chosen are likewise updated. Please refer to the updated manuscript without track changes when reviewing this document.

Description of changes based on journal requirements

Comment 2: Please include a copy if Table xxxxx which you refer to in your text on page 13. 

Response: This has been amended in the updated manuscript. There was a typo in Table numbering in the original manuscript. Line 257 in the original manuscript originally referred to Table 7. “Table 8: Bounds Test” should have read as “Table 7: Bounds Test”. In the updated manuscript, this now reads as Table 6, as the results of the lag length tests have been removed. The updated parts are reproduced here for convenience (Lines 326 and 327): Please see the highlighted parts.

“Table 6 presents the results of the Bounds test for cointegration. Cointegration is strongly suggested at the 1 percent level in the estimated models.”

Table 6: Bounds Test

Model Break Adjusted F-Test Statistic

Tourist arrivals Annual Quarterly

Real exchange rate 32.29A 22.60A

Nominal exchange rate 12.12A 6.80A

Tourism receipts 

Real exchange rate 15.18A 11.08A

Nominal exchange rate 17.24A 6.03A

PCA tourism index 

Real exchange rate 11.46A 22.58A

Nominal exchange rate 6.48A 8.83A

Critical values at 1 percent: I(0) = 4.29; I(1) = 5.61. Model assumes intercept only.

Estimated in EViews 10. A indicates cointegration at the 1 percent level.

Description of changes based on Editors comments

Comment 1: The sample period includes only 24 years of data. The authors should discuss the potential issues associated with the small size of the sample, particularly in the context of the methods used in the paper. Although they comment at the end, they should discuss it much earlier in the text.

Response: We thank the editor for this comment. The potential issues associated with small samples are now discussed in the introduction from lines 79 to 82. Specifically, that small samples may be associated with larger standard errors and smaller effect estimates. In context of the methods used in the paper and whether the methods circumvent the small sample bias, lines 108 to 111 now present a discussion on methods consistent with small samples.

Comment 2: The authors should be careful in discussing the policy implications. Purely based on the results, they can say that any policy promoting tourism in Tonga would contribute to economic growth of the country. The specific policy prescriptions that they have included do not necessarily follow from the data analysis presented in the paper. They are based on the authors' knowledge of the country and its tourism industry. They should clearly acknowledge that.

Response: We thank the editor for this comment. The policy implications section of the conclusion has been duly amended. Please see lines 423 to 446 for both the changes. In lines 423 to 424, we acknowledge that based purely on the results, any policy promoting tourism in Tonga would contribute to economic growth if the country. In lines 426 to 427, we acknowledge that the specific policy prescriptions are based on the authors knowledge of Tonga and its tourism industry.

Comment 3: Although the authors motivate the paper by referring to the potential impacts of Covid-19 on the tourism industry in Tonga, I am afraid there is not much in terms of specific policy intervention in such a situation. First, the dataset does not cover the pandemic period. Second, although the data analysis presents evidence of negative impacts of two earlier events in 2007 and 2010 on growth, it is not clear if they work through the tourism channel. Further, those were destination specific events while Covid-19 has impacted both sources and destination for the tourism industry.

Response: We thank the editor for this comment. The motivation of the paper has been duly amended by excluding the effects of Covid-19. Please see the first 2 paragraphs of the introduction for the updated motivation. We now draw more attention to the benefits and drawbacks of the tourism industry in paragraph 1 and how publication biases may affect policy decisions regarding tourism in paragraph 2. The negative effects of the two identified structural breaks can be justified by the Review article of Nunkoo et al. (2020) which is reference number 3 in the reference list. Tourism is affected by exogenous shocks i.e. exogenous shocks such as economic crisis has an adverse effect on tourism development. 

Please see the end of paragraph 1, section 2 of Nunkoo et al.’s (2020) paper:

Nunkoo R, Seetanah B, Jaffur ZR, Moraghen PG, Sannassee RV. Tourism and economic growth: A meta-regression analysis. Journal of Travel Research. 2020 Mar;59(3):404-23.

We have also updated the discussion of structural breaks under the Section: Materials and Methods (Lines 232 to 237) to emphasize why testing for structural breaks is important. Based on the comment of the editor, we have removed COVID-19 as a motivating factor in this study. 

Comment 4: The null hypotheses stated on p. 9 of the paper are not correct. The null hypothesis of an Augmented Dickey-Fuller test is that the underlying series has a unit root and the rejection of the null hypothesis indicates that the series is stationary. The statement for the BEL test seems to be inconsistent as well.

Response: the null hypothesis of both tests has been duly amended. Please see lines 224 and 238 respectively in the updated manuscript. We thank the editor for this comment.

Comment 5: The results with the tourism indicator (Table 10) seems to be qualitatively similar to those with tourism arrivals (Table 8). The predominance of tourism arrivals in the index (88% of the variations in the index being driven by tourism arrivals) could explain this similarity.

Response: the respective discussion has been updated. Please see lines 342 to 345 in the updated manuscript. Lines 341 to 351 which is the paragraph where this statement occurs is reproduced below for convenience. Please see the highlighted part. 

“The most consistent results are obtained with the PCA tourism development index models (Table 9). The growth effect of tourism is between 0.02 to 0.04, respectively. The results with the tourism indicator (Table 9) are qualitatively similar to those with tourism arrivals (Table 7). The predominance of tourism arrivals in the index which explains 88 percent of the variations in the index could explain this similarity. Unlike earlier results (Tables 7 and 8), the effects are consistent across methods, data frequencies, and exchange rate assumptions. Therefore, developing an overall tourism index is better suited to assess the tourism-growth association. The capital share is between 0.27 to 0.37 which is smaller than those from the earlier estimates (Tables 7 and 8). Consistency is also evident in the exchange rates despite the unit of measurement and is in between 0.08 to 0.12. Overall, the growth effect of tourism is small, positive, and statistically significant in Tonga.”

Description of changes based on reviewer 1 comments:

Comment 1: A review of literature discussing why the author considers the issue of publication bias is necessary. Particularly, the author may provide a summary of how there is a tendency to report favourable and significant results in the tourism-growth literature. This would help supply a conceptual framework to the paper.

Response: We thank the reviewer for this comment. Please see paragraphs 2 to 4 in the updated manuscript introduction for a review of the literature on the publication bias.

Comment 2: The author has selected many methods such as ARDL, DOLS, FMOLS and CCR in the estimation. However, it is not clear why these methods were selected. The suitability of the methods and their likely advantages in the study requires an explanation.

Response: We thank the reviewer for this comment. The suitability and benefits of these methods is discussed in the introduction from lines 106 to 114 and in the updated methodology section from lines 239 to 245 for the ARDL approach and from lines 253 to 260 for the FMOLS, DOLS, and CCR methods. The benefits are generally avoiding the endogeneity bias and providing consistent estimates in small samples, which is the case in our study.

Comment 3: In addition to tourism indicator (tourist arrival, tourist receipt and tourism development index), the author considers only capital stock per worker, exchange rates (nominal and real) and two dummy variables. However, it is unclear why the study is restricted to only these independent variables.

Response: We thank the reviewer for this comment. These independent variables were considered after reviewing the literature on tourism and economic growth and aims to ensure that the estimates comply with theoretical foundations such as the Solow model which is why capital per worker is considered. Exchange rates are included to reduce bias in the estimates. The tourism variables are the main variables of interest. Structural breaks are included because tourism development is affected by exogenous, structural events. Please see paragraph 5, lines 84 to 95 in the introduction for a discussion on this. 

Comment 4: There is a high degree of correlation (presented in table 2) between capital stock and tourist arrival, capital stock and tourist receipt and between nominal exchange rate and tourist arrival. Therefore, it would not be sound to use these variables in the same regression model.

Response: Based on this recommendation, we have added a separate section on ridge regression and presented its estimates as supporting results to the main cointegrating estimates. The methodological discussion on the ridge regression approach is from lines 266 to 287 in the updated manuscript. The ridge estimate results are presented from lines 365 to 373 in the updated manuscript. The respective tests of multicollinearity have also been conducted. The results generally agree with the main estimates from the ARDL, DOLS, FMOLS, and CCR methods. We would like to add that small samples may mimic the effects multicollinearity inflating standard errors and reducing effects sizes. A discussion of the small sample bias is added in the introduction section. We note that the existing methods in our study, the ARDL and DOLS methods provides robust estimates and inferences in small samples according to the following two research papers below. Nevertheless, we duly thank the reviewer for this comment as it allowed us the opportunity to further improve our estimates with a related methodology which was developed to combat issues of multicollinearity and is beginning to be applied in recent studies where it supports the estimates derived from traditional cointegrating models (please see the article by Pan et al., 2021)

Pesaran MH, Shin Y, Smith RJ. Bounds testing approaches to the analysis of level relationships. Journal of applied econometrics. 2001 May;16(3):289-326.

Masih R, Masih AM. Stock-Watson dynamic OLS (DOLS) and error-correction modelling approaches to estimating long-and short-run elasticities in a demand function: new evidence and methodological implications from an application to the demand for coal in mainland China. Energy Economics. 1996 Oct 1;18(4):315-34.

Pan C, Wang H, Guo H, Pan H. How Do the Population Structure Changes of China Affect Carbon Emissions? An Empirical Study Based on Ridge Regression Analysis. Sustainability. 2021 Jan;13(6):3319.

Comment 5: The criteria of lag selection in the causality analysis and unit root test also needs some clarity.

Response: The criteria of the lag length used in the causality and unit root tests has been clarified. Please see the updated subsection on unit roots (lines 208 to 216) and causality (lines 288 to 305) in the updated manuscript for this purpose. For lag length in the augmented Dickey-Fuller test and bandwidth in the Phillips Perron test, we have used the Schwarz information criterion. Please see the updated paragraph below (Lines 208 to 216):

“where ∆ is the first difference operator, y is a time series variable, μ_t is the deterministic component which includes intercept and/or the time trend, φ_1 is the autoregressive coefficient and ϵ_t is the error term. Lagged dependent variables are included in Eq. (4) to correct for potentially auto-correlated residuals under the ADF test. The PP test corrects for auto-correlated residuals through Newey-West standard errors. The lag used in the ADF test and bandwidth in the PP tests is determined by the Schwarz information criteria. The null hypothesis in both tests is that the underlying series has a unit root, H0: φ_1=0. The t-statistics obtained from both methods are compared against the respective critical values. Rejection of the null hypothesis implies that the series in question is stationary.”

The lag length used in the causality analysis is likewise clarified. Please see lines 289 to 295 and lines 375 to 384 in the updated manuscript for a discussion on how the lag length is selected in the VAR model. These are reproduced below for convenience: 

lines 289 to 295 (Please see highlighted parts)

“To examine causality, Toda and Yamamoto’s [39] Granger non-causality test is applied. The advantage of this method is that we can examine causality among variables of a different order of integration, and the method fits well with the ARDL procedure as the part of the information such as lag-length and maximum order of integration is used in the analysis. The maximum lag length is calculated as the sum of the maximum order of integration based on the unit root tests (dmax), and the maximum lag length (l) in the ARDL estimation. In the bivariate case, Toda and Yamamoto’s test VAR is specified as follows:”

lines 375 to 384 (Please see highlighted parts)

“To undertake causality analysis, we rely on the PCA tourism index-real exchange rate model. We set a lag of 1 in the test VAR model which is within the sum of the order of integration and maximum lag of the ARDL model, both which are 1, respectively [39; 41]. The significant causal relations are reported in Table 11 below. Notably, the causality results in Table 11 indicate that tourism, real exchange rates, and capital granger cause growth. Table 11 further indicates that the real exchange rate granger causes tourism and that capital investments granger cause the real exchange rate and tourism, respectively which reaffirms the findings in Table 9. In this regard, predicting Tonga’s economic growth requires a careful analysis of the effect of tourism, real exchange rate, and capital noting the potential inter-relationships. Fig 3 suggests that the test VAR model is stable and hence causality outcomes are reliable.”

Comment 6: The weights used in the calculation of the Tourism Development Index using PCA could be mentioned.

Response: Thank you for this comment. The discussion on the weights/factor loadings of the PCA index has been updated. Please see lines 313 to 317 in the updated manuscript for this purpose. This is reproduced below for convenience: 

“Table 3 below summarizes the PCA results. Noting the vast differences in the mean value of arrivals and receipts, the PCA is run on the log of both indicators. The eigenvalue for arrivals exceeds 1 which indicates its relevance in the index [11]. The factor loading of the first component reveals that arrivals and receipts enter the first component with a similar weight [11]. Around 88 percent of the variation in the tourism index is explained by tourist arrivals.”

Comment 7: Authors should also highlight the key contribution of this research over the existing literature.

Response: Thank you for this comment. The key contribution of this paper has been re-written. Please see lines 135 to 146 in the updated manuscript. The main contribution of the study is the use of multiple triangulation and the development of a framework to remedy publication biases in the tourism-growth literature. This is reproduced below for convenience: 

“The key methodological contribution of this paper is the development of a cohesive framework based on triangulation to remedy publication biases in the tourism-growth literature. The methodology developed draws from the recommendation of Nunkoo et al. [3], theoretical foundations from Song and Wu [10], measurement of tourism from Shahzad et al. [11], and inclusion of moderator variables from Solarin [12]. With this framework, the study provides new evidence on how tourism interacts with growth in small PICs, namely Tonga. Notably, the size and sign of the growth effect of tourism depends on the research method and the measure of tourism. Consistent results across methods and data frequencies are obtained using the overall PCA index models. The findings indicate that tourism has small but positive effects on growth, whilst structural breaks and exchange rates have negative effects on growth. Theoretically consistent values of the capital share are found. The practical implication is on better policies to promote economic growth by developing Tonga’s tourism industry.” 

Description of changes based on reviewer 2 comments:

Comment 1: In P.11, Introduction section, authors talk about the practical implication demands more appropriate policy decisions towards recovery and resilience due to pandemics such as COVID-19. So, in Conclusions section it is expected to argue about policy implications as has been said in Introduction section, but the only sentence can be found in P.25 is “The practical implication is on more appropriate policy decisions for recovery, growth, and resilience through tourism. Enabling investments in basic infrastructures such as roads, airports and international and domestic air transportation, information and communication technology, public amenities, and easing of restrictions to access financial services by tourists would be beneficial. Demand-side factors such as the sensitivity of tourism demand to price and income shocks and the development of a beneficial word of mouth effect need to be re-examined. Tailor-made tourist packages catered for Australian and New Zealand tourists may prove beneficial as would the establishment of direct travel routes.” 

More reflections are needed. Policy recommendations must be highlighted as well as important contributions derived from this work to support one of its main strengths properly.

Certainly, the results indicate that tourism is an important driver of long-run growth in Tonga, like other Pacific islands. However, the paper confirms the overall effect of tourism is smaller than in earlier studies. This may suggest that although the tourism sector influences growth, its magnitude is small relative to competing destinations, and requires further development.

Response: taking the reviewers comment into consideration on further reflections, highlighting the main contribution of the paper, and on the noticeably smaller effect of tourism on growth in Tonga, we have re-worked the policy recommendation paragraphs of the conclusion. Please see lines 414 to 446 in the updated manuscript for this purpose. These are reproduced below for convenience:

“Important contributions derived from the work” (lines 414 to 422):

“Nonetheless, the study could have benefitted from a larger sample size but was restricted to a sample of 1995 to 2018 due to a lack of earlier tourism data. Yet, the key scientific implication/contribution is the development of a cohesive framework that attempts to solve publication biases in the tourism-growth literature. Synthesizing the literature, the framework developed draws from the Solow-Swan growth model, controls for exchange rates and structural breaks, and utilizes an overall indicator of tourism performance. Multiple methods which correct for small sample and endogeneity biases, and multicollinearity are used. Nonlinearity is also considered but found statistically insignificant. Future research can apply the framework advanced in this study to potentially circumvent the publication bias critique.” 

“More reflections are needed. Policy recommendations must be highlighted” (lines 423 to 446)

“Based purely on the results, any policy promoting tourism in Tonga would contribute to economic growth of the country. The practical policy implications need to consider the positive and significant growth effect of tourism, and negative effects arising from political issues and other exogenous shocks. Based on the authors knowledge of Tonga and its tourism industry, policymakers need to make careful decisions in how capital projects are implemented, and how budget shares are allocated to an industry like tourism. This is because resources are limited and there are many other equally urgent competing social projects. To develop the tourism sector, enabling investments in basic infrastructures such as roads, airports and international and domestic air transportation, information and communication technology, public amenities, and easing of restrictions to access financial services is needed. Demand-side factors such as the sensitivity of tourism demand to price and income shocks, and a favourable word of mouth is also important. Tailor-made tourist packages catered for Australian and New Zealand tourists, and the establishment of direct travel routes may also prove beneficial.”

“However, policy decisions to invest in the tourism industry and related areas need to be cautioned based on the noticeably small growth effects found in this study. This implies that although the tourism sector influences growth, its magnitude is small relative to competing destinations, and requires further development. Additionally, the effect of COVID 19 on the tourism industry is unprecedented and requires a radical shift in the way countries depend on tourism [47]. Given the small positive impact of tourism on growth in Tonga, alternative growth strategies such as agriculture, back-office data processing, and call centers that work in tandem with tourism are needed. Further, Tonga needs to pay attention to political stability to avoid the negative effects of tourism on growth. Further research is thus needed to address how much tourism contributes multiplicatively to other industries, like agriculture, and the level of direct and induced employment generated through tourism activities.”

Comment 2: Also, some concerns are related to References along the paper. Here there are some examples but please, revise all:

- P. 9, L. 54: The paper mentions reference “[5,6]” but authors corresponding reference 5 have not been mentioned before.

- P.9, L. 60: It happens the same with “[8]”

- P. 10, L. 76: The same with “[12]”

- P. 10, L. 80: The same with “[14]”

- P. 11, L. 98: The same with “[17]”

- P. 11, L. 100: The same with “[18, 19, 20]”

Response: we thank the reviewer for this comment and apologise sincerely. All references have been thoroughly checked and updated as necessary in the revised manuscript.

---

## [Decision Letter · Decision Letter 1]

31 Aug 2021

PONE-D-21-05441R1

Publication Bias and the Tourism Led Growth Hypothesis

PLOS ONE

Dear Dr. Kumar,

Thank you for submitting your manuscript to PLOS ONE. After careful consideration, we feel that it has merit but does not fully meet PLOS ONE’s publication criteria as it currently stands. Therefore, we invite you to submit a revised version of the manuscript that addresses the points raised during the review process.

Please see my comments below.

We look forward to receiving your revised manuscript.

Kind regards,

Hiranya K. Nath, Ph.D.

Academic Editor

PLOS ONE

Journal Requirements:

Additional Editor Comments (if provided):

The authors have addressed the concerns and questions raised in the first round of review satisfactorily. I would like to accept the paper for publication in PLOS ONE under the condition that the authors address the following issues:

1) p. 6, line 123: delete one of the two “been”s

2) p. 6, line 140: expand PIC as you are using it for the first time

3) p. 7, line 151: use ‘adapted’ instead of ‘adopted’

4) Include data sources in References and cite appropriately

5) p. 9: Give a very brief explanation on why you convert data frequency

6) Include the augmented terms (differenced y lags) in Eq. (4)

Reviewers' comments:

Reviewer's Responses to Questions

**Comments to the Author**

1. If the authors have adequately addressed your comments raised in a previous round of review and you feel that this manuscript is now acceptable for publication, you may indicate that here to bypass the “Comments to the Author” section, enter your conflict of interest statement in the “Confidential to Editor” section, and submit your "Accept" recommendation.

Reviewer #1: All comments have been addressed

2. Is the manuscript technically sound, and do the data support the conclusions?

Reviewer #1: Yes

3. Has the statistical analysis been performed appropriately and rigorously? 

Reviewer #1: Yes

4. Have the authors made all data underlying the findings in their manuscript fully available?

Reviewer #1: Yes

5. Is the manuscript presented in an intelligible fashion and written in standard English?

Reviewer #1: Yes

6. Review Comments to the Author

Reviewer #1: The authors have adequately addressed the comments raised in the previous round. I have no additional comment for the manuscript.

7. PLOS authors have the option to publish the peer review history of their article (what does this mean?). If published, this will include your full peer review and any attached files.

Reviewer #1: No

---

## [Author Response · Author response to Decision Letter 1]

3 Sep 2021

Please see reviewer response document. The responses are reproduced below:

Response to Editors and Reviewers Comments

We would like to thank the handling academic Editor of PLOS ONE, Professor Hiranya Nath, and the anonymous reviewers for accepting the changes we had made in the earlier version of the manuscript. We would also like to thank the journal’s editorial team for efficiently moving the paper through the peer review process. We sincerely appreciate that the reviewers are satisfied with the changes made. The improvements would not have been possible without their useful comments. 

We have checked and updated the references list and ensured that (1) the reference list is complete, and (2) we have not cited any retracted articles in our study consistent with the journals requirements.

We have summarized the changes in a table below in the response document:

Description of Changes

1) p. 6, line 123: delete one of the two “been”s. This has been amended. See line 123 in the updated manuscript

2) p. 6, line 140: expand PIC as you are using it for the first time. This has been amended. The first instance of Pacific Island Countries was in line 125. PIC is defined in line 125 instead of line 140.

3) p. 7, line 151: use ‘adapted’ instead of ‘adopted’. This has been amended. Please see line 151 in the updated manuscript.

4) Include data sources in References and cite appropriately. These are updated and amended. Citations are included from line 185 to 189 in the updated manuscript. The reference section is amended accordingly. These citations are entries 30 to 33. The original citations are amended accordingly. Guidance is sought from the journals reference style guidelines: https://journals.plos.org/plosone/s/submission-guidelines#loc-references

Please see the updated manuscript with track changes to review these changes.

5) p. 9: Give a very brief explanation on why you convert data frequency. This has been updated. Please see lines 197-198 and 205-206 for these changes.

6) Include the augmented terms (differenced y lags) in Eq. (4). Eq. (4) is updated to reflect these changes. Please see line 211 for the updated Eq. (4).

---

## [Editor Report · Decision Letter 2]

5 Oct 2021

Publication Bias and the Tourism Led Growth Hypothesis

PONE-D-21-05441R2

Dear Dr. Kumar,

We’re pleased to inform you that your manuscript has been judged scientifically suitable for publication and will be formally accepted for publication once it meets all outstanding technical requirements.

Kind regards,

Hiranya K. Nath, Ph.D.

Section Editor

PLOS ONE
---

## [Editor Report · Acceptance letter]

7 Oct 2021

PONE-D-21-05441R2 

Publication bias and the tourism-led growth hypothesis 

Dear Dr. Kumar:

I'm pleased to inform you that your manuscript has been deemed suitable for publication in PLOS ONE. Congratulations! Your manuscript is now with our production department. 

Kind regards, 

on behalf of

Dr. Hiranya K. Nath 

Section Editor

PLOS ONE